# Psychometric characteristics of the Hospital Anxiety and Depression Scale in stroke survivors of working age before and after inpatient rehabilitation

Jan Karlsson⊙*, Erik Hammarström, Maria Fogelkvist⊙, Lars-Olov Lundqvist

University Health Care Research Center, Faculty of Medicine and Health, Örebro University, Örebro, Sweden

* jan.karlsson2@regionorebrolan.se

## Abstract

### Objective

The aim was to examine the psychometric properties of the Hospital Anxiety and Depression Scale (HADS) in cohorts of working age stroke survivors, before and after inpatient rehabilitation.

### Methods

Stroke patients aged 18–66 years registered in the national quality register WebRehab Sweden were included in the study at hospital admission (n = 256), discharge (n = 223), and 1-year follow-up (n = 313). Classical and modern (Rasch) methods were used for psychometric evaluation.

### Results

The two-factor HADS model measuring anxiety and depression showed better fit than a single factor measuring emotional distress. The instrument's psychometric stability before and after rehabilitation was satisfactory. The anxiety scale showed good psychometric properties, except for item 7, which is not anxiety-specific. Some concerns were observed for the depression items showing weaker discriminant validity, and item 8 performing poorly as a measure of depression. Cronbach's alpha and McDonald's omega coefficients showed satisfactory internal consistency reliability, whereas Rasch person reliability coefficients indicated weaker reliability, especially for the depression scale. Effect size of change between hospital admission and discharge showed a reduction in anxiety and depression symptoms.

### Conclusions

HADS showed a stable two-factor structure over the rehabilitation period. Patients' perception of items was not affected by the recovery, allowing relevant comparison of HADS scores between different phases of the rehabilitation process. Measures of responsiveness suggest that HADS is sensitive to capturing improvements in emotional distress following

**Data Availability Statement:** Data from Swedish national quality registers in healthcare cannot be shared publicly as disclosure of data is restricted

under Swedish law. Data used in the current study can be requested from the Swedish Register of Rehabilitation Medicine (https://svereh. registercentrum.se) after ethical approval from the Ethical Review Authority (https:// etikprovningsmyndigheten.se/en).

**Funding:** The author(s) received no specific funding for this work.

**Competing interests:** The authors have declared that no competing interests exist.

rehabilitation interventions. Overall, despite minor psychometric weaknesses, HADS is a suitable instrument for assessing anxiety and depresssion symptoms in stroke patients aged 18–66 years.

## Introduction

Stroke is a major cause of disability in the adult population and post-stroke mood disorders are common [1, 2]. About one-third of stroke survivors develop post-stroke depression (PSD), while the prevalence of post-stroke anxiety (PSA) is approximately 20–25% [3, 4]. An increase in emotional distress is observed in the first months after stroke, which may persist and even increase several years post-stroke [5, 6].

Improvement of physical symptoms can be achieved with stroke rehabilitation; however, a corresponding recovery of emotional distress is generally not seen [6]. This may be partly because post-stroke mood disorders remain largely underdiagnosed and untreated, which can have a negative impact on rehabilitation, functional recovery, social interaction, and quality of life [5–9]. By screening for mood disorders, we can identify stroke patients at risk of PSD and PSA and offer them appropriate treatment [5, 8].

Globally, one-third of stroke victims are people of working age; in high-income countries, the proportion is about one-fifth [10]. Stroke can have far-reaching consequences, such as impaired physical, mental, and cognitive function, and rehabilitation efforts are of particular interest to working age patients because they have a long life expectancy. Mood disorders in younger patients have been associated with poorer long-term functional recovery [7], suggesting that screening for PSD and PSA in this age group is important.

The Hospital Anxiety and Depression Scale (HADS) is frequently used in screening for psychological distress in patients with somatic diseases [11, 12]. Advantages of the HADS are that it measures both anxiety and depression, and that it excludes questions about somatic symptoms (e.g., loss of appetite, weight loss, headaches, fatigue, insomnia) that may be caused by somatic illness rather than being an expression of emotional distress. The HADS is easy to administer and has been found to be a valid measure of PSD and PSA, matching other self-rating inventories with regard to sensitivity and specificity [13–17].

Psychometric testing of the HADS has been frequently reported, although the majority of studies have used factor analysis to evaluate its construct validity [11, 12]. In a comprehensive review, Bjelland et al. [11] found that most studies had confirmed the hypothesized two-factor solution of the HADS; however, a more recent review by Cosco et al. [12] concluded that the HADS factor structure was unstable. In particular, they pointed out that the ability of the HADS to distinguish between anxiety and depression constructs was questionable and that the instrument was possibly more useful as a measure of general distress. These findings indicate the importance of evaluating the HADS factor structure in various disease samples to determine its appropriateness for screening purposes.

Over the past decades, the methodological techniques for assessing the psychometric properties of self-report questionnaires have improved, and older instruments like the HADS can be tested with new or enhanced techniques, such as Rasch analysis [18]. Additionally, developments in quality-of-life research have highlighted the importance of evaluating an instrument's psychometric properties across different disease groups to ensure its validity [19]. Few studies have evaluated the psychometric properties of the HADS in stroke samples [20, 21], and studies of working-age stroke patients are lacking.

Our aim was to perform a comprehensive psychometric evaluation using classical test theory (CTT) and modern (Rasch) analysis to test the psychometric properties of the HADS in stroke survivors of working age. A second aim was to evaluate the HADS at different measurement times in order to test the instrument's psychometric stability before and after rehabilitation.

## Methods

### Study sample

Data was obtained from WebRehab Sweden, a national quality register covering about 87% of the Swedish rehabilitation clinics treating patients with acquired brain injury [22]. Stroke is the most common cause of inpatient rehabilitation and includes about 700 patients per year.

Patients in the study were registered between 2013 and 2015 at nine rehabilitation centers. We included patients between 18 and 66 years of age who had answered the HADS at admission (n = 256), and/or at discharge (n = 223), and/or at 1-year follow-up (n = 313) (S1 Table). Patients registered during the same period, who had not answered the HADS, were used for comparison with the above three study cohorts.

Registry data for the current study was accessed on October 12, 2020. The data is de-identified, and the authors did not have access to information that could identify individual participants. Informed consent is obtained when the person is registered in the WebRehab register. Information about the register, how the data is used, and that participation is voluntary, is given both verbally and in writing. By participating in the register, the person agrees that the data can be used for research in health care. No signed written documentation is used for informed consent. The study was approved by the Regional Ethical Review Board in Uppsala, Sweden (Registration number: 2016–055 and 2020–04786).

### Hospital Anxiety and Depression Scale

The HADS consists of 14 items with a 4-point response scale [23, 24]. Item responses are summarized in two scales, one measuring symptoms of anxiety, the other depression. Scores range from 0 to 21 and higher values indicate more emotional distress. Scores are classified as normal (0–7), possible mood disorder (8–10), and probable mood disorder (11–21). A scale score was calculated if at least half of the items in a scale were answered. Missing item values were imputed using a person-specific mean value based on the non-missing items [19].

### Psychometric and statistical analysis

CTT and Rasch analysis were conducted on observed data without imputation for missing item values.

**Internal consistency reliability.**   Cronbach's alpha and McDonald's omega coefficients were computed to estimate the internal consistency reliability [25]. A coefficient $\geq 0.80$ is generally considered desirable; $\geq 0.70$ is considered adequate for group data [26].

**Floor and ceiling effects.**   The proportion scoring at the lowest (floor) and highest (ceiling) possible scale levels was calculated. Floor and ceiling effects were defined as $\geq 15\%$ of the sample scoring at the lowest and highest scale level, respectively [27].

**Construct validity.**   Multitrait scaling analysis was used to test item–scale convergent and discriminant validity [19]. Scaling assumptions were evaluated by calculating Pearson's correlation coefficients between each item and its own subscale (corrected for overlap) and between items and the other scale. A correlation of $\geq 0.40$ is generally considered satisfactory for item–scale convergent validity, while item–discriminant validity is supported if items correlate

higher with the scale it represents than with other scales. The significance of a difference between two item–scale correlations was determined using the standard error of the correlation matrix $(1/\sqrt{n})$. The recommended significance criterion of 2 SE was used [27].

Confirmatory factor analysis (CFA) was used to test the construct validity of a one- and two-factor HADS model. CFA was performed using LISREL 8.8 with generalized least squares estimation on the asymptotic covariance matrices [28]. The PRELIS program was used to obtain the polychoric correlation matrix [29]. Because the data was ordinal, the Satorra–Bentler scaled chi-square (S-B$\chi^2$) was applied [30]. The adequacy of the factor model was evaluated using the comparative fit index (CFI), the Tucker–Lewis index (TLI), the standardized root mean square residual (SRMR), and the root mean square error of approximation (RMSEA). Values $\geq$0.90 and $\geq$0.95 for the CFI and TLI, $\leq$0.10 and $\leq$0.08 for the SRMR, and $\leq$0.08 and $\leq$0.06 for the RMSEA were considered to constitute adequate or excellent goodness of fit [31]. To test whether a one- or two-factor model fitted the data best, S-B$\chi^2$ difference tests were performed using software developed by Crawford and Henry [32].

Known-groups validity was tested based on the hypothesis that HADS scores would be significantly higher for women than for men, and that the effect size (ES) would be small [33]. Significance testing was performed using the Mann–Whitney U-test. Effect size was calculated as the mean group difference divided by the pooled standard deviation (SD) [34]. Criteria for ES are: trivial (0.00–0.19), small (0.20–0.49), medium (0.50–0.79), and large (0.80+).

**Rasch analysis.** The Rasch, one-parameter, partial credit model was used to test the unidimensionality of scales, item fit, item difficulty, response category functioning, reliability, and differential item functioning (DIF). The analyses were performed using WINSTEPS 5.2.0.0 [35].

Unidimensionality, the assumption that items within a scale measure a single latent variable, was tested using principal component analysis of the residuals. The result is considered acceptable if $\geq$50%, and good if $\geq$60%, of the variance can be explained. The eigenvalue of the unexplained variance in the first contrast should be $<2$ [36].

Item fit to the Rasch model was tested using the information-weighted fit statistic (infit) mean square (MnSq) and the outlier-sensitive fit statistic (outfit) MnSq. Infit/outfit values between 0.7 and 1.3 are regarded good quality although a few items with values within the 0.5–1.5 range are considered acceptable [36].

The Rasch one-parameter model estimates item difficulty, which locates the item on the logit scale representing the latent variable. On the HADS, a higher difficulty estimate indicates that the item is "harder" to respond to, i.e., more severe anxiety/depression symptoms are required for endorsement, while a lower estimate indicates that the item is "easier," i.e., less severe anxiety/depression is required for endorsement. An estimate $>0.5$ is considered "hard," between -0.5 and 0.5 denotes "medium," and $<-0.5$ is "easy" [37]. Item maps were produced to illustrate the difficulty/location of each item.

Rasch estimates of person and item reliability and separation were calculated. A reliability coefficient of $\geq$0.85 is regarded as good, 0.80–0.84 as acceptable. A separation index between 2.0 and 2.4 is acceptable and $\geq$2.5 is good [36].

Analysis of the 4-point response categories was performed to test whether the difficulty estimates for each response category increase monotonically, i.e., a more difficult category having a higher logit value. The distance between thresholds is expected to be between 1.4 and 5.0 logits [36]. Category probability curves were used to illustrate the response category functioning.

DIF was tested to evaluate the stability of the item difficulty estimates between the three measurement occasions, at hospital admission, discharge, and 1-year follow-up. The DIF method indicates measurement bias that occurs when individuals at different occasions

respond differently to an item. A DIF contrast $<0.5$ is insignificant, $0.5–1.0$ is mild, and $>1.0$ is notable [36].

**Responsiveness.** Within-group change between admission and discharge was tested with Wilcoxon signed-ranks test. Standardized response mean (SRM) was used to estimate the ES of change. The SRM is calculated as the mean change between assessments, divided by the SD of change [19]. The SRM was judged against standard criteria [34].

## Results

Characteristics of the three study cohorts are shown in Table 1. The majority, 56.9–62.5%, were men. The patients' mean age was 50.8–52.3 years and about two-thirds were between 50 and 66 years old. Roughly half of the patients had an upper secondary education, and about one-third had a university education. Comparisons between HADS respondents and those

**Table 1. Characteristics of the study cohorts of stroke patients who had answered the Hospital Anxiety and Depression Scale (HADS) at admission to inpatient rehabilitation, discharge, and 1-year follow-up, compared with patients who had not answered the HADS.**

| | Admission | | Discharge | | 1-year follow-up | |
|---|---|---|---|---|---|---|
| | HADS | No HADS | HADS | No HADS | HADS | No HADS |
| N | 256 | 1,036 | 223 | 982 | 313 | 397 |
| **Age at admission**, mean (SD) | 50.8 (11.2) | 52.0 (10.2) | 52.3 (10.1) | 51.4 (10.6) | 52.1 (9.6) | 53.8 (9.1) |
| **Age group**, n (%) | | | | | | |
| 18–34 years | 28 (10.9) | 82 (7.9) | 18 (8.1) | 89 (9.1) | 17 (5.4) | 17 (4.3) |
| 35–49 years | 64 (25.0) | 257 (24.8) | 54 (24.2) | 253 (25.8) | 89 (28.4) | 81 (20.4) |
| 50–66 years | 164 (64.1) | 697 (67.3) | 151 (67.7) | 640 (65.2) | 207 (66.1) | 299 (75.3) |
| **Gender**, n (%) | | | | | | |
| Female | 96 (37.5) | 377 (36.4) | 86 (38.6) | 358 (36.5) | 135 (43.1) | 160 (40.3) |
| Male | 160 (62.5) | 659 (63.6) | 137 (61.4) | 624 (63.5) | 178 (56.9) | 237 (59.7) |
| **Educational level**, n (%) | | | | | | |
| Compulsory school | 24 (9.4) | 112 (10.8) | 22 (9.9) | 100 (10.2) | 21 (6.7) | 59 (14.9) |
| Upper secondary school | 122 (47.7) | 485 (46.8) | 109 (48.9) | 458 (46.6) | 161 (51.4) | 207 (52.1) |
| University | 95 (37.1) | 299 (28.9) | 84 (37.7) | 296 (30.1) | 109 (34.8) | 111 (28.0) |
| Other/unknown | 15 (5.9) | 140 (13.5) | 8 (3.6) | 128 (13.0) | 22 (7.0) | 20 (5.0) |
| **Country of birth**, n (%) | | | | | | |
| Sweden | 203 (79.3) | 835 (80.6) | 179 (80.3) | 781 (79.5) | 265 (84.7) | 356 (89.7) |
| Nordic country | 4 (1.6) | 33 (3.2) | 6 (2.7) | 30 (3.1) | 12 (3.8) | 12 (3.0) |
| European country | 12 (4.7) | 63 (6.1) | 16 (7.2) | 57 (5.8) | 15 (4.8) | 5 (1.3) |
| Non-European country | 37 (14.5) | 105 (10.1) | 22 (9.9) | 114 (11.6) | 21 (6.7) | 24 (6.0) |
| **Marital status, children in household**, n (%) | | | | | | |
| Single, no children | 75 (29.3) | 334 (32.2) | 66 (29.6) | 315 (32.1) | 89 (28.4) | 124 (31.2) |
| Single, with children | 19 (7.4) | 54 (5.2) | 20 (9.0) | 57 (5.8) | 20 (6.4) | 20 (5.0) |
| Married, no children | 76 (29.7) | 358 (34.6) | 77 (34.5) | 320 (32.6) | 108 (34.5) | 152 (38.3) |
| Married, with children | 80 (31.3) | 241 (23.3) | 53 (23.8) | 220 (22.4) | 86 (27.5) | 81 (20.4) |
| Other/unknown | 6 (2.4) | 49 (4.6) | 7 (3.1) | 70 (7.1) | 10 (6.7) | 20 (5.0) |
| **Hospital stay**, n (%) | | | | | | |
| Short (1–24 days) | 57 (22.4) | 208 (20.2) | 23 (10.3) | 205 (21.0) | 62 (19.8) | 140 (37.5) |
| Moderate (25–68 days) | 86 (33.7) | 541 (52.5) | 89 (39.9) | 480 (49.3) | 120 (38.3) | 133 (35.7) |
| Long (>68 days) | 112 (43.9) | 281 (27.3) | 111 (49.8) | 289 (29.7) | 131 (41.9) | 100 (26.8) |

HADS = patients who had answered the HADS. No HADS = patients registered during the same period who had not completed the HADS. Married = including cohabiting. Other/unknown = living with parents, another close relative, other arrangement, or unknown. SD = standard deviation.

**Table 2. Descriptive statistics for the Hospital Anxiety and Depression Scale (HADS) for three cohorts of stroke patients at admission to inpatient rehabilitation, discharge, and 1-year follow-up.**

| | Admission (n = 256) | | Discharge (n = 223) | | 1-year follow-up (n = 313) | |
|---|---|---|---|---|---|---|
| | *Anxiety* | *Depression* | *Anxiety* | *Depression* | *Anxiety* | *Depression* |
| Mean | 6.61 | 5.30 | 5.43 | 4.51 | 5.08 | 4.93 |
| SD | 5.04 | 4.11 | 4.54 | 3.88 | 4.23 | 4.13 |
| 95% CI | 5.99–7.23 | 4.79–5.80 | 4.83–6.03 | 4.00–5.02 | 4.61–5.55 | 4.47–5.38 |
| Median | 5.92 | 4.00 | 4.00 | 4.00 | 4.00 | 4.00 |
| Skewness | 0.69 | 0.70 | 0.94 | 1.25 | 0.75 | 0.80 |
| Kurtosis | -0.27 | -0.27 | 0.38 | 1.81 | -0.00 | 0.02 |
| Range | 0–21 | 0–17 | 0–20 | 0–20 | 0–21 | 0–19 |
| Floor, % | 9.0 | 9.0 | 12.6 | 10.8 | 14.4 | 12.1 |
| Ceiling, % | 0.4 | 0.0 | 0.0 | 0.0 | 0.3 | 0.0 |
| Computable scale scores, % | 100 | 100 | 100 | 100 | 100 | 100 |
| Possible cases, % | 14.1 | 14.5 | 13.0 | 14.8 | 14.7 | 16.0 |
| Probable cases, % | 23.1 | 13.3 | 14.8 | 5.8 | 12.1 | 11.2 |

Ceiling, % = proportion of patients scoring at the highest possible scale level. CI = confidence interval. Computable scale scores, % = percentage of patients for whom scale scores were computable. Floor, % = proportion scoring at the lowest possible scale level. Mean = HADS scores range from 0 to 21, with higher values indicating more symptoms. Possible cases, % = proportion scoring between 8 and 10. Probable cases, % = proportion scoring between 11 and 21. SD = standard deviation.

who had not answered the questionnaire showed that the characteristics were roughly similar, suggesting that the study cohorts are representative of Swedish stroke patients aged 18–66 years, referred for inpatient rehabilitation.

## HADS descriptive statistics

The proportion of missing items for both the anxiety and depression scales was low at admission (1.2%) and discharge (0.4–0.8%) and slightly higher at 1-year follow-up (4.1%) (S2 Table). Mean values (range 0–3) for anxiety items were 0.73–1.14 at admission, 0.60–0.95 at discharge, and 0.49–0.90 at 1-year follow-up. The corresponding values for the depression items were 0.61–1.27 at admission, 0.46–1.11 at discharge, and 0.50–1.22 at 1-year follow-up. The highest mean values for items on the anxiety and depression scales were noted for items 11 and 8, respectively.

The proportion of computable scale scores was 100% for both scales (Table 2). At admission, mean (SD) scores were 6.61 (5.04) for anxiety and 5.30 (4.11) for depression, while the corresponding scores at discharge were 5.43 (4.54) and 4.51 (3.88). At 1-year follow-up, mean anxiety and depression scores were 5.08 (4.23) and 4.93 (4.13), respectively.

**Floor and ceiling effects.** The proportion with scores at the lowest and highest possible scale levels was <15% at all three measurement occasions, which means that no floor or ceiling effects were observed (Table 2).

**Possible and probable cases.** At admission, 14.1% and 23.1% were classified as, respectively, possible and probable cases of anxiety disorder (Table 2). The corresponding values for depression were 14.5% and 13.3%. At discharge, the proportion of possible and probable cases was 13.0% and 14.8% for anxiety and 14.8% and 5.8% for depression. At 1-year follow-up, 14.7% and 12.1% were classified as possible and probable cases of anxiety disorders, while 16.0% and 11.2% exceeded the cut-off level for probable or possible depression.

## Construct validity

**Multitrait scaling analysis.** At admission, the correlations between items and the own scale varied from 0.63 to 0.77 for the anxiety scale and from 0.49 to 0.67 for the depression scale (S3 Table). Correlations at discharge were 0.60–0.76 for anxiety and 0.45–0.75 for depression. At 1-year follow-up, correlations ranged between 0.62 and 0.73 for anxiety and between 0.53 and 0.78 for depression. All item–scale correlations showed satisfactory item-convergent validity (r ≥0.40).

Correlations between items and the other scale demonstrated several scaling errors. For the anxiety scale, the correlation for item 7 at admission, and for item 7 and 9 at discharge and follow-up did not meet the criterion for item-discriminant validity. For the depression scale, correlations for items 4, 6, 8, and 14 at admission, items 6, 8, 10, and 14 at discharge, and items 8, 10, and 14 at 1-year follow-up indicated weak item-discriminant validity.

**Interscale correlations.** Strong correlations between the anxiety and depression scales were observed at admission (r = 0.66), discharge (r = 0.67), and 1-year follow-up (r = 0.73).

**Confirmatory factor analysis.** CFA of the two-factor HADS model showed RMSEA values <0.06, CFI and TLI values >0.95, and SRMR values ≤0.083, indicating excellent model fit, while worse fit was obtained for the one-factor model (Table 3). The S-B$\chi^2$ difference tests showed better fit for the two-factor model at admission ($\Delta$S-B$\chi^2$ = -33.83, p<0.001), discharge ($\Delta$S-B$\chi^2$ = -35.56, p<0.001), and 1-year follow-up ($\Delta$S-B$\chi^2$ = -50.98, p<0.001). Factor loadings are shown in S4 Table.

**Known-groups validity.** The mean anxiety and depression scores at admission were significantly higher for women than for men (S5 Table). At discharge and 1-year follow-up, the anxiety scores were significantly higher for women, while the depression scores were roughly equal. The ES of the gender difference was small for anxiety at all measurement times, while the ES for depression was small at admission and trivial at discharge and follow-up.

## Rasch analysis

Tests of unidimensionality for a single HADS factor showed that the explained variance was below the 50% criterion at two measurements, and the eigenvalues of the first contrast were higher than the recommended value at all three measurements (Table 3). Corresponding tests for the anxiety scale showed that the explained variance was acceptable at all three measurements (59.3%, 57.2%, and 54.7%). Results for the depression scale were acceptable at admission (52.9%) and discharge (58.4%), but just below the recommended value at 1-year follow-up (49.5%). Eigenvalues of the unexplained variance in the first contrast met the recommended criterion for both scales.

Infit and outfit statistics showed good fit to the Rasch model for all anxiety items and good or acceptable fit for the depression items, except for one misfit value for depression item 8 (outfit = 1.64 logits) at 1-year follow-up (Table 4).

Tests of the 4-point response scale showed that the distance between threshold values was within the recommended range for three anxiety items (7, 11, 13) at all assessments, while values below the desirable range were noted for four anxiety items (1, 3, 5, 9). However, these deviations were noted at only one of the three measurement occasions (S6 Table, S1 Fig). For the depression items, deviations from the recommended range were observed for all items at admission, for six items at discharge, and for one item at follow-up (S7 Table, S2 Fig). Values below the appropriate range were noted for threshold estimates between response category 2 and 3. Disordered difficulty values for categories 2 and 3 were observed for three depression items (2, 6, 14). However, fewer than ten observations were noted for category 3 on several items.

**Table 3. Results of confirmatory factor analysis (CFA) of a one- and two-factor Hospital Anxiety and Depression Scale (HADS) model, and Rasch analysis of unidimensionality, in stroke patients at admission, discharge, and 1-year follow-up.**

| | Admission (n = 256) | Discharge (n = 223) | 1-year follow-up (n = 313) |
|---|---|---|---|
| **CFA, one-factor model** | | | |
| df | 77 | 77 | 77 |
| S-B$\chi^2$ | 278.745*** | 320.563*** | 356.684*** |
| Normal theory $\chi^2$ | 658.550*** | 863.256*** | 939.139*** |
| RMSEA | 0.101 | 0.119 | 0.108 |
| RMSEA 90% CI | 0.0887–0.114 | 0.106–0.133 | 0.0967–0.119 |
| CFI | 0.968 | 0.958 | 0.971 |
| TLI | 0.968 | 0.958 | 0.971 |
| SRMR | 0.084 | 0.093 | 0.068 |
| **CFA, two-factor model** | | | |
| df | 76 | 76 | 76 |
| S-B$\chi^2$ | 99.367* | 113.519*** | 121.196*** |
| Normal theory $\chi^2$ | 253.140*** | 331.706*** | 342.117*** |
| RMSEA | 0.035 | 0.047 | 0.044 |
| RMSEA 90% CI | 0.009–0.052 | 0.028–0.065 | 0.028–0.058 |
| CFI | 0.996 | 0.994 | 0.995 |
| TLI | 0.996 | 0.994 | 0.995 |
| SRMR | 0.055 | 0.083 | 0.057 |
| **Rasch analysis** | | | |
| *Total (14 items)* | | | |
| Unidimensionality | | | |
| Variance explained | **48.3%** | **48.8%** | 51.9% |
| Eigenvalue | **2.59** | **2.75** | **2.39** |
| *Anxiety* | | | |
| Unidimensionality | | | |
| Variance explained | 59.3% | 57.2% | 54.7% |
| Eigenvalue | 1.50 | 1.47 | 1.54 |
| *Depression* | | | |
| Unidimensionality | | | |
| Variance explained | 52.9% | 58.4% | **49.5%** |
| Eigenvalue | 1.50 | 1.53 | 1.45 |

CFI = comparative fit index. CI = confidence interval. df = degrees of freedom. RMSEA = root mean square error of approximation. S-B$\chi^2$ = Satorra–Bentler scaled chi-square

(*p<0.05

**p<0.01

***p<0.001).

SRMR = standardized root mean square residual. TLI = Tucker–Lewis index/non-normed fit index. Rasch analysis = partial credit model; variance explained by the first factor (acceptable value ≥50%); eigenvalue of the unexplained variance in the first contrast (acceptable value <2). Values **in bold** do not meet the recommended criteria.

The difficulty or item location estimates (in logits) for the anxiety items ranged between -0.69 and 0.84 at the three assessments; the corresponding values for the depression items were -1.76–0.99 (Table 4, Fig 1). Four anxiety items (1, 3, 5, 7) and three depression items (2, 12, 14) had difficulty estimates in the "medium" range (-0.5–0.5 logits) at all three measurements. Two anxiety items (9, 13) were in the "hard" range (logits >0.5) at two measurements and one item (11) was in the "easy" range (logits <-0.5) at all three measurements. Of the depression

**Table 4. Difficulty estimates (logits) and fit statistics for the anxiety and depression items of the Hospital Anxiety and Depression Scale (HADS), according to the Rasch partial credit model, at admission, discharge, and 1-year follow-up.**

| Items | Admission (n = 256) | | | Discharge (n = 223) | | | 1-year follow-up (n = 313) | | |
|---|---|---|---|---|---|---|---|---|---|
| | Diff. | Infit | Outfit | Diff. | Infit | Outfit | Diff. | Infit | Outfit |
| *Anxiety* | | | | | | | | | |
| 1. I feel tense or "wound up." | 0.17 | 0.90 | 0.89 | -0.21 | 0.84 | 0.83 | -0.35 | 0.84 | 0.84 |
| 3. I get a sort of frightened feeling as if something awful is about to happen. | 0.01 | 0.94 | 0.93 | 0.07 | 1.10 | 1.08 | -0.05 | 1.11 | 1.07 |
| 5. Worrying thoughts go through my mind. | -0.47 | 0.99 | 1.00 | -0.31 | 0.82 | 0.82 | -0.34 | 0.89 | 0.86 |
| 7. I can sit at ease and feel relaxed. | 0.05 | 1.21 | 1.24 | -0.03 | 1.23 | 1.24 | -0.10 | 1.13 | 1.13 |
| 9. I get a sort of frightened feeling, like "butterflies" in the stomach. | 0.22 | 0.90 | 0.91 | 0.59 | 0.98 | 1.04 | 0.84 | 0.88 | 0.84 |
| 11. I feel restless as if I have to be on the move. | -0.58 | 1.26 | 1.24 | -0.56 | 1.10 | 1.13 | -0.69 | 1.23 | 1.23 |
| 13. I get sudden feelings of panic. | 0.60 | 0.76 | 0.74 | 0.46 | 0.82 | 0.80 | 0.68 | 0.86 | 1.00 |
| *Depression* | | | | | | | | | |
| 2. I still enjoy the things I used to enjoy. | 0.19 | 1.05 | 0.98 | 0.48 | 0.86 | 0.74 | 0.30 | 0.93 | 0.88 |
| 4. I can laugh and see the funny side of things. | 0.49 | 0.86 | 0.90 | 0.96 | 0.80 | 0.71 | -0.50 | 0.78 | 0.79 |
| 6. I feel cheerful. | 0.33 | 0.85 | 0.79 | 0.05 | 0.91 | 0.93 | 0.55 | 0.89 | 0.86 |
| 8. I feel as if I'm slowed down. | -1.36 | 1.24 | 1.26 | -1.45 | *1.40* | *1.40* | -1.76 | *1.39* | **1.64** |
| 10. I have lost interest in my appearance. | 0.44 | 1.02 | 1.03 | 0.40 | 1.24 | *1.42* | 0.99 | 1.02 | 1.13 |
| 12. I look forward with enjoyment to things. | -0.08 | 0.85 | 0.81 | -0.30 | 0.73 | 0.69 | 0.01 | 0.72 | 0.70 |
| 14. I can enjoy a good book or radio or TV program. | -0.01 | 1.10 | 1.20 | -0.14 | 1.03 | 1.06 | 0.40 | 1.21 | 1.13 |

Diff = difficulty estimate (logits), where >0.5 = "hard," -0.5–0.5 = "medium," and <-0.5 = "easy." Infit = information-weighted fit statistic, mean square.

Outfit = outlier-sensitive fit statistic, mean square. Infit/outfit values of 0.7–1.3 indicate good quality; a few items between 0.5 and 1.5 are acceptable. Fit values outside 0.7–1.3 but within the 0.5–1.5 range are indicated *in italics*; values outside the acceptable range are given **in bold**.

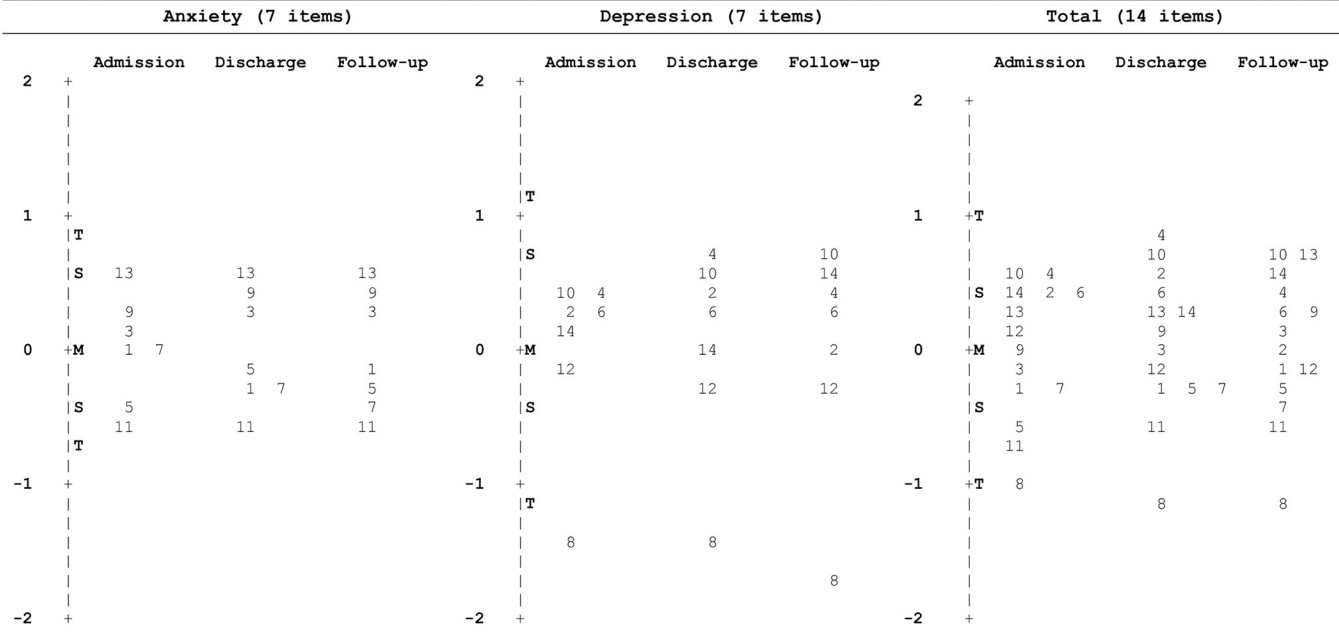

**Fig 1. Item map of the anxiety and depression subscales and the total scale of the Hospital Anxiety and Depression Scale (HADS), showing difficulty/location on the latent variable, at admission, discharge, and 1-year follow-up.** A higher placement on the logit scale indicates a more difficult item. M = mean; S = 1 standard deviation (SD) from the mean; T = 2 SDs from the mean.

items, three items (4, 6, 10) were in the "hard" range on one occasion and in the "medium" range at the other two measurements. Depression item 8 had difficulty estimates between -1.36 and -1.76, indicating a "very easy" item.

**Differential item functioning.** Tests of DIF between the measurements at admission, discharge, and 1-year follow-up showed no DIF for the anxiety items, while mild DIF was noted for depression item 2 at two measurements (DIF contrast = 0.50 and 0.58 logits) (S8 Table).

## Reliability

The internal consistency reliability coefficients (Cronbach's alpha and McDonald's omega) and item reliability and separation estimates for the anxiety and depression scales were above the recommended minimum value at all measurements (Table 5). Person reliability and separation values were slightly below the recommended value for the anxiety scale at discharge and follow-up, while the corresponding estimates for the depression scale were clearly below the acceptable level at all three measurement times.

## Responsiveness

A total of 147 (57.4%) patients completed the HADS at both admission and discharge. In this subgroup, the mean (SD) score change at discharge was -1.24 (3.89) on the anxiety scale and -1.11 (3.37) on the depression scale (p<0.001). The ES of change was 0.32 for anxiety and 0.33

**Table 5. Reliability estimates of the anxiety and depression subscales and the total scale of the Hospital Anxiety and Depression Scale (HADS), at admission, discharge, and 1-year follow-up.**

| Reliability estimates | Admission (n = 256) | Discharge (n = 223) | 1-year follow-up (n = 313) |
|---|---|---|---|
| *Anxiety (7 items)* | | | |
| Cronbach's alpha | 0.90 | 0.89 | 0.89 |
| McDonald's omega | 0.90 | 0.89 | 0.89 |
| Person reliability | 0.83 | **0.79** | **0.79** |
| Item reliability | 0.90 | 0.88 | 0.95 |
| Person separation, logits | 2.19 | **1.93** | **1.94** |
| Item separation, logits | 3.02 | 2.68 | 4.19 |
| *Depression (7 items)* | | | |
| Cronbach's alpha | 0.83 | 0.85 | 0.88 |
| McDonald's omega | 0.83 | 0.85 | 0.88 |
| Person reliability | **0.71** | **0.70** | **0.77** |
| Item reliability | 0.97 | 0.96 | 0.98 |
| Person separation, logits | **1.58** | **1.54** | **1.83** |
| Item separation, logits | 5.28 | 5.14 | 6.92 |
| *Total HADS (14 items)* | | | |
| Cronbach's alpha | 0.91 | 0.91 | 0.92 |
| McDonald's omega | 0.91 | 0.92 | 0.93 |
| Person reliability | 0.84 | 0.83 | 0.85 |
| Item reliability | 0.95 | 0.94 | 0.96 |
| Person separation, logits | 2.33 | 2.21 | 2.39 |
| Item separation, logits | 4.32 | 4.07 | 5.25 |

Cronbach's alpha and McDonald's omega ≥0.70 is acceptable for group data. For person and item reliability ("real" estimate), 0.80–0.84 is acceptable, and ≥0.85 is good. For person and item separation ("real" estimate), between 2.0 and 2.4 is acceptable, and ≥2.5 is good. Values outside the acceptable range are given **in bold**.

for depression, indicating a small improvement on both scales. Anxiety and depression scores at discharge had improved for 59.2% and 52.4% of patients, respectively, while 18.4% and 20.4% were unchanged, and 22.4% and 27.2% had deteriorated.

## Discussion

The study evaluated the psychometric properties of the HADS in cohorts of stroke patients aged 18–66 years who had been referred to inpatient rehabilitation. The HADS was tested using CTT and Rasch one-parameter analysis at three different time points, namely at hospital admission, discharge, and 1-year follow-up.

Completeness of data was satisfactory, indicating that the questionnaire was well accepted by the respondents. No floor or ceiling effects were noted, suggesting that the HADS has satisfactory sensitivity for measuring anxiety and depression symptoms in the lower as well as the upper part of the latent variables in stroke survivors. Multitrait scaling analysis showed satisfactory item–scale convergent validity. However, weak item–scale discriminant validity was noted for two anxiety and five depression items. Poor discriminant validity for anxiety item 7 has been previously reported because this item tends to have roughly equal loadings on both the anxiety and the depression factor [11]. However, discriminant validity for the depression items has not been evaluated in sufficient detail in previous studies. Weak discriminant validity may indicate that the items capture general emotional distress rather than depressive symptoms.

Correlations between the anxiety and depression scales were quite strong, and since half of the 14 HADS items exhibited overlap between the two latent variables, this may suggest that a single latent variable measuring general emotional distress should be used instead of, or as a complement to, the two-factor model. Previous studies have shown conflicting results regarding the HADS measurement model, and one- to four-factor models have been proposed, although best fit for the two-factor model is the most common finding [11, 12]. In the current study, CFA indicated excellent fit for the two-factor model, whereas the one-factor model exhibited poorer fit indices. Rasch analysis of unidimensionality showed that separate measures of anxiety and depression explained a greater proportion of variance than did the one-factor model. Additionally, the eigenvalues of the unexplained variance met the recommended criterion for both anxiety and depression, while eigenvalues for the one-factor model indicated more than one factor. In conclusion, tests of construct validity showed that a two-factor model is most appropriate for interpretation of HADS data in stroke survivors, which is in line with two previous studies [20, 21].

Item fit to the Rasch model was evaluated by infit and outfit statistics. All anxiety items and five depression items exhibited good fit at all measurement occasions. Depression item 10 showed acceptable to good fit, while a value above the recommended range was noted for item 8 ("I feel as if I'm slowed down"). A too high infit or outfit value may indicate that the item is not measuring the same latent variable defined by the rest of the items.

Difficulty estimates for item 8 indicated that it was "very easy," and it tended to fall outside the difficulty range of all other items, which were mostly in the "medium" range. Previous studies of patients with various medical conditions have shown that item 8 functions poorly [38–40]. Patient responses appear to be related to the consequences of somatic illness rather than to the degree of emotional distress, leading to slight overestimation of depressive symptoms [20, 21, 41].

Response category functioning was tested to determine the extent to which the four categories used to rate the items are chosen in a logical order. The results indicate that the respondents can meaningfully distinguish the four steps on all anxiety items, while problems with

disordered difficulty estimates between categories 2 and 3 were noted for three depression items. However, fewer than ten observations for category 3 were noted for several items, indicating that the study samples were too small to reliably evaluate this scale step.

Analysis of DIF was performed to test the stability of HADS item difficulty or item location estimates between measurements. No DIF was observed for anxiety items, while mild DIF was noted for depression item 2. Item 2 ("I still enjoy the things I used to enjoy") is the only retrospective HADS item, which may affect the response pattern at different measurement times before and after rehabilitation; however, the cause of DIF needs to be investigated in qualitative studies. Overall, HADS items were sufficiently stable, meaning that the instrument can be used to assess symptoms of anxiety and depression at different times during rehabilitation. Analysis of DIF by gender, age, education, or clinical parameters was not performed in the current study, but such analyses may provide additional insights into the performance of HADS items [42].

Cronbach's alpha and McDonald's omega coefficients demonstrated satisfactory internal consistency reliability, consistent with previous studies [11, 21]. Rasch estimates of item reliability and separation also met the recommended requirements. Sufficient item separation values indicate that items are reasonably separated on the logit scale representing the latent variable. However, Rasch person reliability and separation estimates for the depression scale were below the adequate level, suggesting that the measure's ability to discriminate between respondents can be improved. Rasch person reliability coefficients for both scales were lower than alpha and omega values, giving rise to inconsistent interpretations. However, the Rasch measure is considered to provide a more correct estimate [43].

The ES of change between hospital admission and discharge showed a small reduction in both anxiety and depression symptoms, suggesting that the instrument is sensitive to capturing improvements in emotional distress following rehabilitation interventions.

Two previous studies have evaluated the psychometric properties of the HADS in stroke survivors [20, 21]. However, these studies included older patients and comparison with our samples is therefore uncertain.

A strength of the study is that the HADS was evaluated using both CTT and Rasch analysis. Different analysis techniques can complement each other and provide a broader basis for interpretation, which can strengthen the evaluation of an instrument's psychometric properties and contribute to more reliable conclusions [18]. Another strength is that the HADS was evaluated at three different measurement times pre- and post-rehabilitation. Psychometric evaluations are usually performed on data from one measurement, but assessment instruments such as the HADS should provide valid scores even after interventions, which was tested in this study. The results show that deviant results can occur at one measurement, but that similar patterns are not replicated on all measurement occasions. If similar results are obtained at several measurements, this indicates that the findings are reliable, while single deviations at one measurement should be interpreted with some caution.

The study has some limitations. The sample sizes were mainly sufficient for psychometric testing, but the observations for response category 3 were too small to reliably analyze the response category functioning. The study used samples from a national registry and the generalizability of our findings may be somewhat uncertain. However, comparisons of patients with and without HADS data showed that characteristics at hospital admission were roughly similar, suggesting that the study cohorts are representative of Swedish stroke patients aged 18–66 years, referred for inpatient rehabilitation. The recommended diagnostic thresholds for the HADS have been questioned and lower cut-offs have been suggested for screening stroke patients [13–16]; however, this issue was not evaluated in the study.

## Conclusion

The two-factor HADS model measuring anxiety and depression showed better fit than a single factor measuring emotional distress. The instrument's psychometric stability before and after rehabilitation was satisfactory, indicating that the patients' perception of items was not affected by the recovery, allowing relevant comparisons between different phases of the rehabilitation process. The anxiety scale generally showed good psychometric properties except for item 7, which is not anxiety-specific and tends to measure general emotional distress. Some concerns were noted for the depression items, which showed weaker discriminant validity, and responses to depression item 8 seemed to be related to somatic consequences of stroke rather than to mood disorders. In summary, however, the overall psychometric results suggest that the HADS is suitable for assessing anxiety and depression symptoms in working age stroke patients.

## Supporting information

**S1 Fig. Category probability curves showing response category functioning for the anxiety scale of the Hospital Anxiety and Depression Scale (HADS) according to the Rasch partial credit model, at admission, discharge, and 1-year follow-up.**
(DOCX)

**S2 Fig. Category probability curves showing response category functioning for the depression scale of the Hospital Anxiety and Depression Scale (HADS) according to the Rasch partial credit model, at admission, discharge, and 1-year follow-up.**
(DOCX)

**S1 Table. Stroke patients included in the analysis, at hospital admission to inpatient rehabilitation (cohort 1), at discharge (cohort 2), and at 1-year follow-up (cohort 3).**
(DOCX)

**S2 Table. Descriptive statistics for the anxiety and depression items of the Hospital Anxiety and Depression Scale (HADS) among stroke patients at admission to inpatient rehabilitation, at discharge, and at 1-year follow-up.**
(DOCX)

**S3 Table. Multitrait scaling analysis of the Hospital Anxiety and Depression Scale (HADS) items at admission, discharge, and 1-year follow-up.**
(DOCX)

**S4 Table. Confirmatory factor analysis (CFA) loadings of the Hospital Anxiety and Depression Scale (HADS) items for one- and two-factor models at admission, discharge, and 1-year follow-up.**
(DOCX)

**S5 Table. Known-groups validity tests of male–female differences on the Hospital Anxiety and Depression Scale (HADS) at admission, discharge, and 1-year follow-up.**
(DOCX)

**S6 Table. Response category functioning (Andrich threshold estimates) for Hospital Anxiety and Depression Scale (HADS) anxiety items, according to the Rasch partial credit model, at admission, discharge, and 1-year follow-up.**
(DOCX)

**S7 Table. Response category functioning (Andrich threshold estimates) for Hospital Anxiety and Depression Scale (HADS) depression items, according to the Rasch partial credit model at admission, discharge, and 1-year follow-up.**
(DOCX)

**S8 Table. Rasch analysis of differential item functioning (DIF) for Hospital Anxiety and Depression Scale (HADS) items between the measurement occasions (at admission, discharge, and 1-year follow-up).**
(DOCX)

## Author Contributions

**Conceptualization:** Jan Karlsson, Erik Hammarström, Maria Fogelkvist, Lars-Olov Lundqvist.

**Data curation:** Maria Fogelkvist, Lars-Olov Lundqvist.

**Formal analysis:** Jan Karlsson, Erik Hammarström, Maria Fogelkvist, Lars-Olov Lundqvist.

**Methodology:** Jan Karlsson, Lars-Olov Lundqvist.

**Project administration:** Jan Karlsson, Lars-Olov Lundqvist.

**Supervision:** Jan Karlsson, Lars-Olov Lundqvist.

**Writing – original draft:** Jan Karlsson.

**Writing – review & editing:** Jan Karlsson, Erik Hammarström, Maria Fogelkvist, Lars-Olov Lundqvist.

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
