## [Decision Letter · Decision Letter 0]

28 Mar 2024

PONE-D-24-05048Psychometric characteristics of the Hospital Anxiety and Depression Scale in stroke survivors of working age before and after inpatient rehabilitationPLOS ONE

Dear Dr. Karlsson,

Thank you for submitting your manuscript to PLOS ONE. After careful consideration, we feel that it has merit but does not fully meet PLOS ONE’s publication criteria as it currently stands. Therefore, we invite you to submit a revised version of the manuscript that addresses the points raised during the review process.

As the editor I want to tell you that, in my opinion, your paper is of high quality. Nevertheless, it might improve if you considered the comments of the reviewers seriously, either by adopting them or by pointing out why you don't. Moreover, I am not sure whether all literature recommended by the reviewers must necessarily cited. It is up to you to judge this.

We look forward to receiving your revised manuscript.

Kind regards,

Uwe Konerding

Academic Editor

PLOS ONE

2.  note that you have indicated that there are restrictions to data sharing for this study. PLOS only allows data to be available upon request if there are legal or ethical restrictions on sharing data publicly. For more information on unacceptable data access restrictions, please see http://journals.plos.org/plosone/s/data-availability#loc-unacceptable-data-access-restrictions. 

3. Please include a copy of Table 6 which you refer to in your text on page 26.

Reviewers' comments:

Reviewer's Responses to Questions

**Comments to the Author**

1. Is the manuscript technically sound, and do the data support the conclusions?

Reviewer #1: Yes

Reviewer #2: Yes

2. Has the statistical analysis been performed appropriately and rigorously? 

Reviewer #1: Yes

Reviewer #2: Yes

3. Have the authors made all data underlying the findings in their manuscript fully available?

Reviewer #1: Yes

Reviewer #2: Yes

4. Is the manuscript presented in an intelligible fashion and written in standard English?

Reviewer #1: Yes

Reviewer #2: Yes

5. Review Comments to the Author

Reviewer #1: In this study the authors examined the psychometric properties of the HADS in cohorts of working age stroke survivors. The study has an interesting content. The analyses were done with great care. Nevertheless, I think the manuscript can still be improved by consider the comments raised below:

Major general comment:

* Cisco and colleagues (2012) were able to show in their article, based on a 10-year systematic review, that the HADS has an inability to differentiate between the two constructs anxiety and depression. Coyne and van Sonderen wrote provocatively in 2012 "No further research needed: Abandoning the HADS". On the one hand, I would encourage the authors to justify a little more why they did another analysis (based on this facts/hypotheses). Secondly, I would like to see more reference of their results to other studies/analyses on HADS.

Further comments:

Abstract:

The use of subheadings (such as Objects, Methods, Results, ...) could improve readability.

Possibly remove the double mention of the aspects around items 7 and 8 and replace them with another aspect.

Introduction:

P. 4: The statement that "the majority of studies..." seems to be correct. I don’t know the whole literature. However, I would expect at least somewhere in the article (here or in the discussion) a reference/comparison to the existing analyses with Rasch (e.g. Tang et al. 2007, or Ayis et al. 2018).

P. 4: Analyses based on both CCT and IRT are performed. A explanation for this is given in the discussion. Possibly paste here.

Methods:

P. 5: An imputation of missing values is performed. If I understand the results correctly, then the missing rate was very low. Would it perhaps have been an option to only calculate with already complete data sets? It would be worth considering whether a sentence could be added to the limitations pointing out possible difficulties with imputations.

p. 8: I understand the use of "harder/easier" here and in the rest of the article. The Rasch Model has been used in many analyses in the educational field and there this use of language makes a lot of sense. However, I am not sure if it makes as much sense in the clinical field. A change/adaptation should be considered.

p. 8: Only DIF analyses to the measurement times were calculated. It should be considered whether further analyses would be useful, e.g. on gender, age or a clinical parameters.

Results:

p. 9: The title includes "survivors of working age". Is there no data on the patients' occupational status?

p. 10: "Most anxiety symptoms were reported on item 11 ...". I find the wording a bit problematic, especially for item 8. With this statement, we assume that the items measure the underlying construct in a valid way. Your analyses were not able to show this, at least for item 8. I would find it more favorable if, for example, you chose a formulation such as "on item 11 were found the hightest mean values" (just a suggestion).

p. 11: Where do the groupings 8-10 and 11-21 come from? Is this the information from the manual? If so, please cite the source.

p. 12: "possible and probable cases": Reading this gives the impression that a high value indicates a certain disorder. Especially in the area of anxiety, it's not so simple. There is not "one" anxiety disorder. A high score can be an indication of an agoraphopia, a social phobia, a trauma disorder or another disorder.

Discussion:

p. 20: "measuring mood disorders" - see comment above.

p. 23: "Different analysis techniques.... " Please provide a source for this statement.

Reviewer #2: Manuscript Title:

Psychometric characteristics of the Hospital Anxiety and Depression Scale in stroke survivors of working age before and after inpatient rehabilitation.

ID: PONE-D-24-05048

Comments:

1. The introduction is clear, overviewing the main concepts and making the basic case well for the current study.

2. I found the results to be explained clearly and thoroughly. Tables/Figures support the written results.

3. Methods overall are clearly described, including analytic procedures.

4. Author/s should provide the value of MacDonald's Omega as an index of reliability because up-to-date research suggest that Alpha might not be interpreted as an internal consistency coefficient. I suggest that Cronbach's alpha should not be used as a measure of internal consistency. See

Sijtsma, K. (2009). On the use, the misuse, and the very limited usefulness of Cronbach’s alpha. psychometrika, 74, 107-120.‏ https://doi.org/10.1007/s11336-008-9101-0

5. For the Item Response Theory (Rasch analysis) and MacDonald's Omega please cite the following article:

Khalaf, M.A., Omara, E.M.N. Rasch analysis and differential item functioning of English language anxiety scale (ELAS) across sex in Egyptian context. BMC Psychol 10, 242 (2022). https://doi.org/10.1186/s40359-022-00955-w

6. For anxiety and depression literature, please cite the following article:

Khalaf, M. A., & Al-Said, T. T. (2021). The Egyptian validation study of the resilience scale for adults (RSA) and its utility in predicting depression. The Open Psychology Journal, 14, 83-92. https://doi.org/10.2174/1874350102114010083

7. Do not use abbreviations in the abstract.

8. Author/s wrote: " frequently used in screening for 60 psychological distress in patients with somatic diseases". You should provide more than 3 references as long as you said, "frequently used".

10. You wrote: " Patients in the study were registered between 2013 and 2015 at nine rehabilitation centers". There is a wide gap between the year of data collection and the year of publishing 2024.

11. You should provide the psychometric properties of The HADS that are reported in its original study.

12. For anxiety, please cite the following article:

Khalaf, M.A., Shehata, A. M. (2023). Trust in information sources as a moderator of the impact of COVID-19 anxiety and exposure to information on conspiracy thinking and misinformation beliefs: a multilevel study. BMC Psychology, 11, 375. https://doi.org/10.1186/s40359-023-01425-7

13. You should provide the differential item functioning (DIF) of the items.

Finally, My decision is: minor correction

6. PLOS authors have the option to publish the peer review history of their article (what does this mean?). If published, this will include your full peer review and any attached files.

Reviewer #1: No

Reviewer #2: No

---

## [Author Response · Author response to Decision Letter 0]

22 May 2024

Please see our comments in the "Response to Reviewers" document.

---

## [Decision Letter · Decision Letter 1]

9 Jun 2024

PONE-D-24-05048R1Psychometric characteristics of the Hospital Anxiety and Depression Scale in stroke survivors of working age before and after inpatient rehabilitationPLOS ONE

Dear Dr. Karlsson,

Thank you for submitting your manuscript to PLOS ONE. After careful consideration, we feel that it has merit but does not fully meet PLOS ONE’s publication criteria as it currently stands. Therefore, we invite you to submit a revised version of the manuscript that addresses the points raised during the review process.

I think that following the suggestions of reviewer 1 would improve the manuscript. So, I would be glad if you would follow them.

We look forward to receiving your revised manuscript.

Kind regards,

Uwe Konerding

Academic Editor

PLOS ONE

Journal Requirements:

Reviewers' comments:

Reviewer's Responses to Questions

**Comments to the Author**

1. If the authors have adequately addressed your comments raised in a previous round of review and you feel that this manuscript is now acceptable for publication, you may indicate that here to bypass the “Comments to the Author” section, enter your conflict of interest statement in the “Confidential to Editor” section, and submit your "Accept" recommendation.

Reviewer #1: (No Response)

Reviewer #2: (No Response)

2. Is the manuscript technically sound, and do the data support the conclusions?

Reviewer #1: Yes

Reviewer #2: Partly

3. Has the statistical analysis been performed appropriately and rigorously? 

Reviewer #1: Yes

Reviewer #2: Yes

4. Have the authors made all data underlying the findings in their manuscript fully available?

Reviewer #1: Yes

Reviewer #2: Yes

5. Is the manuscript presented in an intelligible fashion and written in standard English?

Reviewer #1: Yes

Reviewer #2: No

6. Review Comments to the Author

Reviewer #1: Reviewer #1 _ Revision 2.0

Dear author, thank you very much for the comprehensive answers and the changes that have already been implemented. There are currently two areas that remain open for me, for which I hope a simple solution/compromise can be found.

1.) For the most part, I can support your comments. And I am a great advocate of further psychometric studies. I have read and commented on your manuscript and remain convinced that the introduction is a little short on existing analyses of the HADS and the need for this study. You described this very clearly in our brief exchange - but I feel this is somewhat lacking in the manuscript. My suggestion would therefore be that you add 4-5 sentences in the introduction. I think that there are readers (like me) who would like to have a little more information about the state of research and your reasons for the study.

Comment 1.0: “Cisco and colleagues (2012) were able to show in their article, based on a 10-year systematic review, that the HADS has an inability to differentiate between the two constructs anxiety and depression. Coyne and van Sonderen wrote provocatively in 2012 "No further research needed: Abandoning the HADS". On the one hand, I would encourage the authors to justify a little more why they did another analysis (based on this facts/hypotheses). Secondly, I would like to see more reference of their results to other studies/analyses on HADS.”

Respond: “In a comprehensive review of 71 papers, Bjelland et al (2002) concluded that "HADS was found to perform well in assessing symptom severity and incidence of anxiety disorders and depression in both somatic, psychiatric and primary care patients and in the general population." This finding provides support for the HADS as a useful instrument for assessing anxiety and depression symptoms in diverse samples. While we acknowledge opposing opinions, such as those of Coyne and van Sonderen, we believe that there are significantly more studies supporting the validity of the HADS. Regarding the HADS factor structure, Bjelland et al (2002) concluded that “Most factor analyses demonstrated a two-factor solution in good accordance with the HADS subscales forAnxiety (HADS-A) and Depression (HADS-D), respectively.” However, a more recent review by Cosco et al (2012) found that “Twenty-five of the 50 reviewed studies revealed a two-factor structure, the most commonly found HADS structure. Additionally, five studies revealed unidimensional, 17 studies revealed three-factor, and two studies revealed four factor structures.” This finding indicates the importance of evaluating the HADS factor structure in different samples to ascertain its appropriateness. This is one reason why we evaluated the factor structure in the current study. However, it should be noted that studies obtaining 3-factor solutions most often present different subfactors for anxiety. These subfactors are highly correlated, and it is questionable whether the division offers any clinical advantages. If the aim is to screen for anxiety symptoms, we believe it is sufficient to measure anxiety as one factor, as confirmed in the current study. There are several other reasons to continue testing the psychometric properties of the HADS. Firstly, the methodological techniques for assessing the psychometric properties of self-report questionnaires have been improved over the past decades. Older instruments like the HADS can be tested with new or improved techniques, such as Rasch analysis. Secondly, developments in quality-of-life research have highlighted the importance of evaluating an instrument's psychometric properties across different disease groups to ensure its appropriateness. Thirdly, to enhance the measurement of various domains, it is crucial to study individual items to identify areas for improvement within the instrument. We believe the motives for the current study are adequately described in the Introduction, and no further additions are necessary.”

2.) The second area concerns the DIF analyses. It is perfectly understandable to focus and I am not asking for additional analyses in this manuscript. However, I would like you to add a sentence that further analyses on DIF (test fairness) are possible and can provide further insights. The sentence can simply be added somewhere in the discussion. There are some studies (not on stroke patients, I assume), that may offer certain hypotheses for possible DIF problems (gender, age, etc.) (e.g. Cameron et al. 2013).

Comment 1.0: “Only DIF analyses to the measurement times were calculated. It should be considered whether further analyses would be useful, e.g. on gender, age or a clinical parameters.”

Respond: “Thank you for your comment. A key aspect highlighted in our study's findings is the conceptual consistency of HADS over time, i.e., no DIF was observed. This indicates that patients maintain a stable perception of the questions' meaning, unaffected by the rehabilitation efforts. However, we believe that potential DIF concerning factors like gender or age falls beyond the scope of our current investigation, but remains a broader question for the HADS instrument as a whole.”

Reviewer #2: Some of my comments were not Done at all. I advise authors to do them. I think they will increase the quality of the manuscript.

7. PLOS authors have the option to publish the peer review history of their article (what does this mean?). If published, this will include your full peer review and any attached files.

Reviewer #1: No

Reviewer #2: No

---

## [Author Response · Author response to Decision Letter 1]

18 Jun 2024

Please see our responses in the document Response to Reviewers

---

## [Editor Report · Decision Letter 2]

24 Jun 2024

Psychometric characteristics of the Hospital Anxiety and Depression Scale in stroke survivors of working age before and after inpatient rehabilitation

PONE-D-24-05048R2

Dear Dr. Karlsson,

We’re pleased to inform you that your manuscript has been judged scientifically suitable for publication and will be formally accepted for publication once it meets all outstanding technical requirements.

Kind regards,

Uwe Konerding

Academic Editor

PLOS ONE
---

## [Editor Report · Acceptance letter]

27 Jun 2024

PONE-D-24-05048R2 

PLOS ONE

Dear Dr. Karlsson, 

I'm pleased to inform you that your manuscript has been deemed suitable for publication in PLOS ONE. Congratulations! Your manuscript is now being handed over to our production team.

Kind regards, 

on behalf of

Dr. Uwe Konerding 

Academic Editor

PLOS ONE